# Modeling of Magnetospheres of Terrestrial Exoplanets in the Habitable Zone around G-Type Stars

Elena S. Belenkaya *[ID], Igor I. Alexeev [ID] and Marina S. Blokhina

Skobeltsyn Institute of Nuclear Physics (SINP MSU), Federal State Budget Educational Institution of Higher Education M.V. Lomonosov Moscow State University, 1(2), Leninskie Gory, GSP-1, 119991 Moscow, Russia; alexeev@dec1.sinp.msu.ru (I.I.A.); vyazanka@bk.ru (M.S.B.)

* Correspondence: elena@dec1.sinp.msu.ru

**Abstract:** Using a paraboloid model of an Earth-like exoplanetary magnetospheric magnetic field, developed from a model of the Earth, we investigate the magnetospheric structure of planets located in the habitable zone around G-type stars. Different directions of the stellar wind magnetic field are considered and the corresponding variations in the magnetospheric structure are obtained. It is shown that the exoplanetary environment significantly depends on stellar wind magnetic field orientation and that the parameters of magnetospheric current systems depend on the distance to the stand-off magnetopause point.

**Keywords:** exoplanet; magnetosphere; stellar wind; magnetic field; G-type stars





## 1. Introduction

Exoplanetary atmospheres are controlled by the stellar winds and the radiation from their host stars. Stellar winds have existed since the beginning of stellar evolution [1]. The interaction of stellar winds and their magnetic fields with exoplanets is very important for the possibility of life on planets. As a first step in this study, exoplanets from habitable zones (HZ) are considered. In the habitable zone, it is assumed that water can exist on the surface in liquid form. To maintain this condition in particular, a stable atmosphere is needed. The boundaries of the HZ are determined by loss of water and by condensation. We pay attention to the exoplanetary magnetic field, which is very important for the atmosphere and the existence of life. The planetary magnetic field protects the atmosphere from erosion with strong stellar winds. The influence of the stellar wind magnetic field on an exoplanet's environment was considered, in particular, by Belenkaya et al. [2] and Khodachenko et al. [3].

Stellar winds interacting with exoplanets modify their environment. The cavity around exoplanets, in which the planetary magnetic field prevails over the stellar wind magnetic field and determines plasma behavior, is called the magnetosphere. The boundary of the magnetosphere is the magnetopause. The pressure from stellar wind compresses the magnetopause forward, and an extended magnetotail appears on the opposite side. The interaction of stellar wind with the planetary magnetosphere leads to the formation of magnetospheric current systems, the magnetic field of which contributes to the total magnetospheric field.

Stellar winds have been studied in a series of papers. For example, MacLeod and Oklopčić [4] modeled the interaction between planetary and stellar winds. Cassinelli [5] reviewed theories of stellar winds from luminous stars of early and late types. High-velocity stellar winds from super-giant stars (type O and B) were detected. Vink [6] noted that massive stars are a danger to nearby exoplanets because their stellar winds bring a lot of mass and energy. Johnstone et al. [7] estimate that stellar wind properties for main sequence stars have masses close to that of the sun.

The magnetic field of stellar wind plays a main role in planetary–star interaction (or IMF—interplanetary magnetic field). Unfortunately, at present, our knowledge of magnetized stellar winds is insufficient. Most of this knowledge comes from solar wind study. It is for this reason that research on exoplanets regarding solar-like stars is very active. For example, Reda et al. [8] considered the correlation of solar wind variability with the sun's activity during five solar cycles. The authors noted that this study is important, not only for the solar system, but also for other solar-type stars in their interactions between the stellar wind and exoplanets. The analogy of stellar wind from sun-type stars with solar wind was used by Nichols and Milan [9] in their study of auroral radio emission due to the stellar wind–exoplanet's magnetosphere interaction. In the work [10], using 3D MHD models, the authors calculated the Earth's magnetospheric stand-off distance during the sun's evolution along the main sequence, dependent on the decreasing solar rotation speed. G-class stars of the Hertzprung–Russell diagram, to which the sun belongs, are yellow with surface temperatures of $5 \times 10^3$–$6 \times 10^3$ K. They are X-ray sources. Their spectra contain characteristic lines of iron, sodium, calcium, magnesium, and titanium. Their masses in solar masses are from 0.8 to 1.04, and their sizes are from 0.96 to 1.15 sun radii. The main term in the stellar wind pressure for G stars is the ram pressure, while for M stars it is the magnetic pressure [11,12]. Magnetic field perturbations in stellar wind affect planetary aurora, atmospheres, radio emission, reconstruct the magnetosphere, and can modify the HZ [13–20].

The aim of the paper is to develop the paraboloid model of the terrestrial-like planetary magnetosphere in the habitable zone around a sun-type star. The influence of the stellar wind magnetic field on exoplanetary magnetospheres is of decisive importance. Using a paraboloid magnetospheric exoplanetary model, we show how this effect is realized. Section 2 describes the paraboloid model of the exoplanet's magnetospheric magnetic field. Numerical results are presented in Section 3. Section 4 contains conclusions.

## 2. Model Description

The paraboloid magnetospheric magnetic field model was developed based on the terrestrial model by Alexeev [21]. This model includes its own planetary magnetic field, the fields of magnetospheric current systems: the magnetotail and the magnetopause. The partially penetrated magnetic field from the stellar wind is also considered. The coefficient of IMF penetration into the magnetosphere, $k$, is not known. It may be taken ~0.2, as it is on Earth (for a detailed discussion of this issue, see [22]). The IMF value was arbitrarily chosen to be 100 nT.

In the model, the shape of the magnetopause is a paraboloid of revolution:

$$x/r_{mp} = 1 - (y^2 + z^2)/2r_{mp}^2.$$

The calculations are carried out in the stellar-magnetospheric coordinate system, where the X axis is directed from the planet's center to the star; the Z axis is perpendicular to X in the plane containing the X and the planetary dipole axes; Y points to dusk perpendicular to X and Z. Here, we ignore the tilt angle between Z and the magnetic dipole axis; $r_{mp}$ is the planetocentric distance to the nose of the magnetopause. In the terminator plane a magnetopause cross-section radius is equal to $(2)^{1/2} r_{mp}$ or at x = 0 and y = 0, z = 1.4 $r_{mp}$ (see Figure 1).

The axisymmetric shape of the magnetopause is probably due to the symmetry of the supersonic flow around a blunt obstacle (the magnetosphere). Since we do not know the conditions on the fictitious exoplanet, only a very rough model can be used. For this reason, we did not take into account the possible asymmetry of the magnetopause shape associated with the magnetospheric magnetic field asymmetry and assumed an axisymmetric shape for the magnetopause. Experimental data confirm this assumption for Earth [23], Jupiter [24], and Mercury [25].

The planetary magnetic field is a dipole with a value $B_0$ ~$3.1 \times 10^4$ nT at the equator (as on Earth). The magnetopause current shields all magnetospheric magnetic field sources.

The protection of an exoplanet works better if the magnetospheric scale (which is the stand-off distance at the magnetopause, $r_{mp}$) is much larger than the exobase height [3].

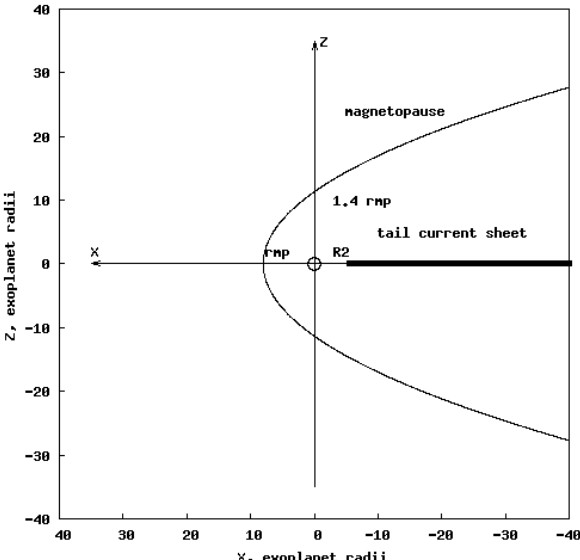

**Figure 1.** The scheme illustrates used parameters in the paraboloid model.

The magnetotail current system consists of cross currents in the equatorial neutral sheet and closure currents at the magnetopause. The distance from the planetary center to the inner edge of the magnetotail current sheet is $R_2$. The magnetic field of the tail current system there is characterized by the parameter $B_t$. This parameter linearly depends on the flux in the tail lobe $F_{lobe} = B_t \pi r_{mp}^2 / 2$ [26].

A detailed description of magnetospheric current systems calculated in the paraboloid magnetic field model is given, for example, in Belenkaya et al. [27] for Jupiter and Alexeev et al. [28] for Mercury. The main model parameters used here are: (1) the magnetopause stand-off distance from the exoplanet center, $r_{mp}$; (2) distance from the exoplanet center to the inner edge of the tail current sheet, $R_2$; (3) magnetic field of the tail current system at the inner edge of the tail current sheet, $B_t$; (4) components of the stellar wind magnetic field vector (IMF) in the stellar-magnetospheric coordinate system; (5) coefficient of IMF penetration into the magnetosphere, $k$, chosen equal to 0.2; (6) the magnetic field at the exoplanet equator, $B_0$ and (7) the radius of the exoplanet, $R_{pl}$.

The total dipole field plus magnetopause screened current field is considered as curl-free inside the magnetosphere (between the exoplanet and the magnetopause). The scalar potential of the magnetic field of the magnetopause current in the parabolic coordinates was calculated using Bessel functions of the first kind and modified Bessel functions with a singularity at the origin. This is the solution of the Laplace equation for the Neumann problem using integral representations.

The magnetic field of the tail current system is considered as a combination of two parts: the field from the cross-tail current in the infinitely thin current sheet and the field from the closure currents at the magnetopause. For calculations in parabolic coordinates, the magnetic field components were expanded into a mathematical series using Bessel functions of the first kind, modified Bessel functions with singularity at the origin, and the modified Bessel functions without singularity in the magnetosphere.

## 3. Numerical Results

The equation for $r_{mp}$ for solar type stars can be obtained from the equality of the ram stellar wind pressure $P_{sw}$ and the magnetospheric magnetic field pressure $B_m^2/2\mu_0$, if we assume that, as for the Earth, the plasma pressure in the exoplanet's magnetosphere is low.

$$P_{sw} = B_m^2/2\mu_0 \qquad (1)$$

Here, $\mu_0$ is the magnetic permeability of the vacuum.

See et al. [20] considered the non-spherical shape of the magnetopause by introducing the coefficient $f_0 \sim 1.16$ in the right side of Equation (1). Here, we neglect it, because the value of the planetary dipole is known with a large degree of uncertainty, and this would be an overshoot. For this reason, in the rough estimation, we also do not take into account the Chapmen–Ferroro currents, which increase the magnetospheric magnetic field at the stand-off magnetopause point by 2.4 times. Thus, at the stand-off point $B_m = B_0 R_{pl}^3/r_{mp}^3$, where $B_0$ is magnetic field at the planet's equator and $R_{pl}$ is the exoplanet's radius.

$$P_{sw} = (B_0 R_{pl}^3/r_{mp}^3)^2/2\mu_0. \qquad (2)$$

$$2\mu_0 P_{sw} = (B_0 R_{pl}^3/r_{mp}^3)^2 \text{ or } 2\mu_0 P_{sw} r_{mp}^6 = B_0^2 R_{pl}^6 = M_{pl}^2 \qquad (3)$$

where $M_{pl}$ is the exoplanet's magnetic moment.

$$r_{mp} = [B_0^2 R_{pl}^6/(2\mu_0 P_{sw})]^{1/6}. \qquad (4)$$

See et al. [20] studied 124 sun-like stars with exoplanets in the HZ zone. Their masses were from 0.8 to 1.4 solar masses. The authors suggested that around them in the HZ there are fictitious exoplanets with the same masses, sizes, and magnetic fields as those of the Earth. They determined the ram stellar wind pressure, which allows the formation of magnetospheres comparable in size to the Earth's. The Parker [29] and Cranmer and Saar [30] models for the stellar wind and star's mass loss rate, respectively, were used. See et al. [20] calculated the distance from the planet's center to the front point of the magnetopause $r_{mp}$. For the selected 10 stars, they determined that $r_{mp}$ for fictitious Earth-type exoplanets were from 8 to 16 $R_E$, where $R_E$ is the Earth's radius $R_E = 6400$ km. Following See et al. [20] we take the magnetic dipole field of a fictitious terrestrial exoplanet equal to the Earth's with a magnetic moment $M_E = 3.1 \times 10^4$ nT$\cdot R_E^3$. Here, we determine the paraboloid magnetospheric magnetic field model parameters for both values of $r_{mp}$ (8 and 16 $R_{pl}$).

Following See et al. [20], we consider the minimum and maximum values of $r_{mp}$ for 10 selected fictitious planets around solar-like stars and determine the corresponding $P_{sw}$ from Equation (2). We can determine the magnetic field in the tail lobes $B_{lobe}$ from the equality of $P_{sw-th}$ to the $B_{lobe}^2/2\mu_0$, where $P_{sw-th}$ is the thermal stellar wind pressure. Since we do not know exactly the stellar wind parameters, we replace $P_{sw-th}$ here by $P_{sw}$, which is indeed much larger than $P_{sw-th}$. This will lead to a compression of the tail diameter, which, in some sense, compensates for its anomalous increase in the paraboloid model. The magnetic flux in the tail lobe equals to $F_{lobe} = B_{lobe} S_{lobe}$, where $S_{lobe} = (\pi/2) R_{lobe}^2$. We suppose that $R_{lobe} = 1.4\, r_{mp}$; $R_{lobe}^2 = (1.4)^2\, r_{mp}^2 = 1.96\, r_{mp}^2 \sim 2\, r_{mp}^2$.

$$F_{lobe} = B_{lobe} S_{lobe} = B_{lobe}(\pi/2)\, R_{lobe}^2 = \pi B_{lobe}\, r_{mp}^2 \qquad (5)$$

$$P_{sw} = B_{lobe}^2/2\mu_0. \qquad (6)$$

$$B_{lobe}^2 = 2\mu_0\, P_{sw}.$$

$$F_{lobe} = \pi\, (2\mu_0\, P_{sw})^{1/2}\, r_{mp}^2$$

The parameter $B_t$ determines the magnetic field at the inner edge of the magnetotail current sheet located at a distance from the planet's center $R_2$. We assume that

$$R_2 = 0.7\, r_{mp}, \tag{7}$$

as on Earth. From Johnson et al. [26], for a tail current with zero thickness in the paraboloid model, we obtain:

$$F_{lobe} = 0.5 B_t \pi r_{mp}{}^2$$

$$B_t = 2F_{lobe}/(\pi r_{mp}{}^2) \tag{8}$$

Using Equation (4) we receive:

$$B_t = 2F_{lobe}/(\pi r_{mp}{}^2) = 2\pi (2\mu_0 P_{sw})^{1/2}\, r_{mp}{}^2/(\pi r_{mp}{}^2) = 2((B_0 R_{pl}{}^3/r_{mp}{}^3)^2)^{1/2} = 2B_0 R_{pl}{}^3/r_{mp}{}^3$$

$$B_t = 2B_0 R_{pl}{}^3/r_{mp}{}^3 \tag{9}$$

From Equations (7) and (9) we receive from the selected values of $r_{mp}$:

**(a)** $r_{mp} = 8R_E$

$$R_2 = 0.7\, r_{mp} = 5.6\, R_{pl}$$

$$B_t = 2B_0 R_{pl}{}^3/r_{mp}{}^3 = 2B_0/8^3 = 2 \times 3.1 \times 10^4\ nT/8^3 = 6.2 \times 10^4\ nT/512 = 620 \times 10^2\ nT/512 = 121\ nT$$

**(b)** $r_{mp} = 16\, R_E$

$$R_2 = 0.7\, r_{mp} = 11.2\, R_{pl}$$

$$B_t = 2B_0 R_{pl}{}^3/r_{mp}{}^3 = 620 \times 10^2\ nT/16^3 = 620 \times 10^2\ nT/4096 = 15\ nT.$$

Thus, the stand-off distance $r_{mp}$ is determined by the stellar wind dynamic pressure and the magnetospheric magnetic field, the main input to which gives the planetary dipole in the case of an Earth-like exoplanet around a G-type star. Parameter $R_2$ is taken to be proportional to $r_{mp}$ with a coefficient of ~0.7. $B_t$ depends on the planetary dipole moment $M_{pl} = B_0 R_{pl}{}^3$ and $r_{mp}$. Following See et al. [20], we took the $r_{mp}$ values, which they found; the other parameters turned out to be determined by the above equations. Figure 2 presents the results of the calculations.

Figure 2 shows how IMF modifies the magnetospheric magnetic field structure, and how it changes dependent on the $r_{mp}$ value. The structure of the magnetic field when changing the distance to the front point of the magnetopause, $r_{mp}$, does not change in a similar way due to the non-linear dependence of $B_t$ (the magnetic field of the tail current system at the inner edge of the neutral current sheet) on this parameter.

As the magnetosphere is expended ($r_{mp}$ is greater), the value of the parameter $B_t$, which determines the magnetic field of the tail current system, decreases, and the influence of the tail becomes smaller. For this case, the high IMF strength ~100 nT (relative to Earth's environment), which controls the magnetospheric structure, becomes more noticeable. Figure 2b,f show that when the stellar wind magnetic field is parallel to the planetary dipole axis, the tail length for small $r_{mp}$ is ~40 $R_{pl}$, while for large $r_{mp}$ it is ~25 $R_{pl}$. When the IMF is antiparallel to the dipole of an exoplanet (Figure 2c,g), we have a closed magnetosphere without open field lines, which is located for $r_{mp} = 8\ R_{pl}$ from 8 to −34 $R_{pl}$, and for $r_{mp} = 16\ R_{pl}$ from 16 to −20 $R_{pl}$. When IMF = 0 (Figure 2a,e), we have open field lines that do not intersect the magnetopause but travel to the distant tail. For strong azimuthal and radial IMF, we see in Figure 2d,h significant asymmetry between the northern and southern hemispheres, which increases for the expanded magnetosphere. These results are new, despite the fact that the effects connected with the stand-off distance for Earth have been studied in many papers [31–35].

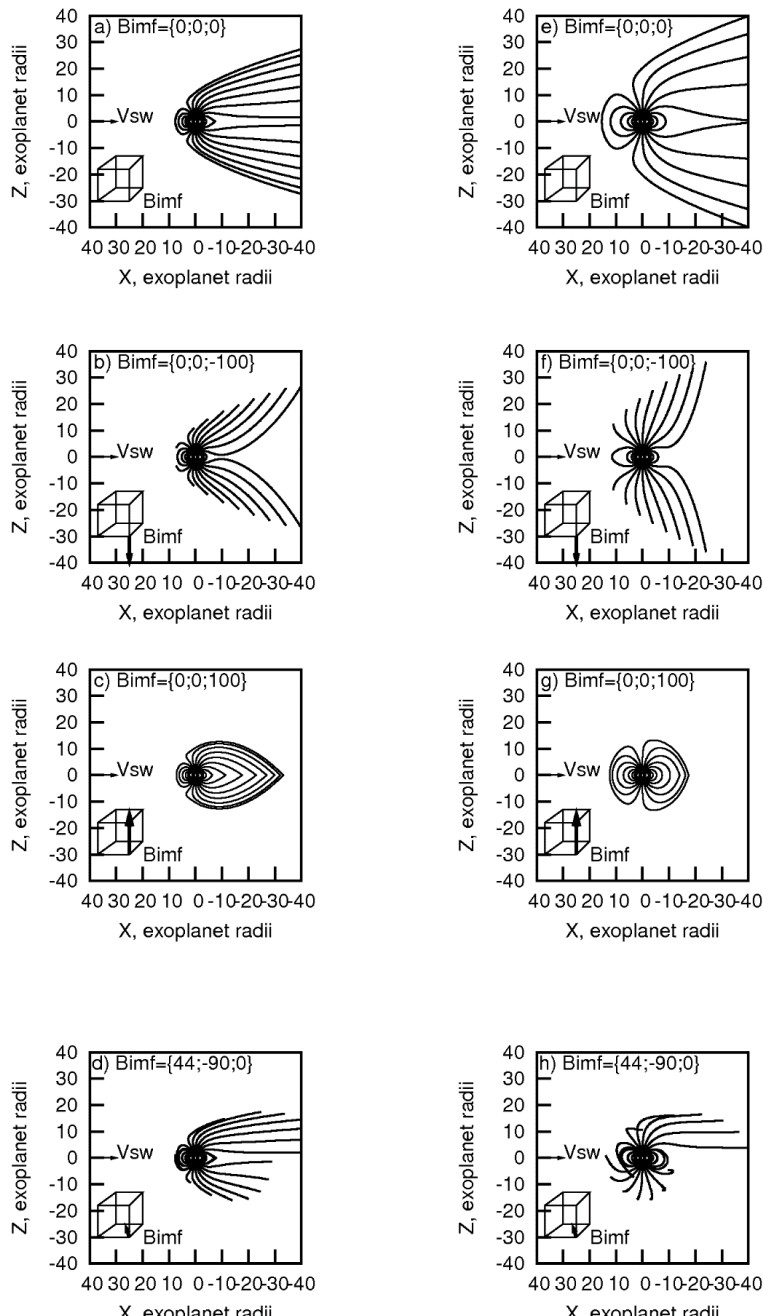

**Figure 2.** The figure shows variations of magnetospheric magnetic field structure for rmp = 8 Rpl, R2 = 5.6 Rpl, Bt = 121 nT (left column; subfigures (**a**–**d**)) and rmp = 16 Rpl, R2 = 11.2 Rpl, Bt = 15 nT (right column; subfigures (**e**–**h**)) for different orientations of the stellar wind magnetic field (IMF). The components of the IMF are given for each image. The coefficient of IMF penetration *k* = 0.2. Magnetic field lines projected on the plane XZ are plotted. Vector $V_{sw}$ marks the average stellar wind velocity.

## 4. Conclusions

In this paper, we considered Earth-like exoplanets in the HZ around solar-type stars using a paraboloid model of the magnetospheric magnetic field. Two sets of parameters were considered, corresponding to the minimum and maximum values of the magnetopause stand-off distance determined by See et al. [20] for ten selected fictitious terrestrial planets around sun-type stars. We have considered the influence of the stellar wind magnetic field of different orientations on the magnetospheric magnetic field structure. For this, a paraboloid model of the magnetospheric magnetic field was used. It turns out that the

magnetospheric structure significantly depends not only on the orientation of the IMF, but also on the stand-off distance $r_{mp}$, since all the parameters that determine magnetospheric current systems are controlled by it. Particularly sensitive to $r_{mp}$ is the magnetic field of the tail current system at the inner edge of the neutral current sheet, $B_t$, which is inversely proportional to $r_{mp}{}^3$.

For zero stellar wind magnetic field, open field lines do not intersect magnetopause but travel to the distant tail. For IMF antiparallel to the exoplanet's dipole, the closed magnetosphere without open field lines arises. For the considered stellar wind magnetic field strength ~100 nT, the size of such closed magnetosphere is from 8 to −34 $R_{pl}$ for $r_{mp} = 8 R_{pl}$ and from 16 to −20 $R_{pl}$ for $r_{mp} = 16 R_{pl}$. For IMF parallel to the exoplanet's dipole, the tail length for $r_{mp} = 8 R_{pl}$ is ~40 $R_{pl}$, while for $r_{mp} = 16 R_{pl}$ it is ~25 $R_{pl}$. For strong azimuthal and/or radial IMF, the significant asymmetry between the northern and southern hemispheres arises, which increases for the expanded magnetosphere.

A decrease in the $B_t$ value with the growth of $r_{mp}$ leads to significant reconstruction of the magnetospheric magnetic field. Such a situation was not considered in detail for Earth, despite that the variations in $r_{mp}$ are similar according to the work of Pudovkin et al. [36], for example, who stated that the experimental data suggest the change in the Earth's stand-off distance from 6.6 to 13.7 terrestrial radii.

Moreover, we consider the stellar wind magnetic field with the value of 100 nT, which does not exist on Earth. It is shown that the influence of such a strong stellar wind magnetic field on the magnetospheric structure increases with the growth of $r_{mp}$.

**Author Contributions:** Conceptualization, E.S.B. and I.I.A.; methodology, E.S.B., I.I.A., M.S.B.; software, I.I.A., M.S.B.; validation, I.I.A.; formal analysis, E.S.B., I.I.A., M.S.B.; investigation, E.S.B., I.I.A.; writing—original draft preparation, E.S.B.; writing—review and editing, E.S.B., I.I.A.; visualization, M.S.B.; supervision, E.S.B.; project administration, E.S.B.; All authors have read and agreed to the published version of the manuscript.

**Funding:** This research received no external funding.

**Institutional Review Board Statement:** Not applicable.

**Informed Consent Statement:** Not applicable.

**Data Availability Statement:** Not applicable.

**Acknowledgments:** Authors acknowledge the support of Ministry of Science and Higher Education of the Russian Federation under the grant 075-15-2020-780 (N13.1902.21.0039).

**Conflicts of Interest:** The authors declare no conflict of interest.

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
