# Peer review of "Modeling of Magnetospheres of Terrestrial Exoplanets in the Habitable Zone around G-Type Stars"

_universe, doi:10.3390/universe8040231_

Round 1

Reviewer 1 Report

The manuscript is devoted to theory along with calculations to study the important parameters of the magnetosphere of Earth-like exoplanet. This is a modern topic possibly related to the important universe experiments. The authors have many years of experience in the study of planetary magnetospheres in the solar system. The manuscript is well written and can be recommended for publication in the journal “Universe” after revision taking into account the comments below.

1) In the section “Numerical results”, it is necessary to clearly indicate which parameters of the problem under consideration are considered to be given in the calculations and why. It is necessary to specify the parameters, which can be very different due to lack of data. As a result of the section, it is necessary to formulate specific restrictions on the expected characteristics of the exoplanet's magnetosphere.

2) Figure 1 needs to be presented in a better quality. At the pictures are useful to show the orientation and magnitude of the magnetic field and the speed of the stellar wind.

3) In the text, it is necessary to formulate the results that follow from the gallery of pictures presented in Figure 1.

4) In conclusion, it is not necessary to discuss occasionally the possibility of life on an exoplanet without a serious study of this issue in the text.

Author Response

Please, see the attached file

Reviewer 2 Report

The authors describe a numerical model of the magnetosphere of an Earth-like exoplanet under varying stellar wind IMF conditions.

I have some concerns regarding the paper that I am hoping the authors can address:

  • Line 40-43: Here the authors describe the lack of knowledge of stellar winds outside our solar system, and so most of the knowledge is applied from our own solar wind. Some references to this, very widely studied phenomenon would be useful.
  • Line 49-51: Here the authors discuss pertubations in the stellar wind affecting aurora, atmosphere etc...Again, references would be useful.
  • Section 2 - Model description: A diagram describing the geometry could be helpful here, especially to coincide with figure 1 presented later.
  • Figure 1 - It is impossible to read any of the detail on Figure 1. This needs to be addressed as a matter of urgency, as it is a fundamental aspect of the paper.
  • related to figure 1 - There is no significant discussion about any of the results presented in figure 1. What are the implications, and what conclusions do the authors draw from these results?
  • Line 159-162: Here the authors discuss the magnetosphere structure being affected by the orientation of the IMF and the standoff distance, Rmp.

At this point, I question the novelty of the paper. Earth's magnetosphere is subject to the solar wind, with the direction of the IMF and the magnetopause stand-off distance affecting the dynamics of the aurora, substorms, radio emissions, reconnection events, etc....yet there are few references anywhere in the paper to the decades of research performed on our own magnetosphere. What work have the authors done that has not already been applied to Earth?

Author Response

Please, see the attached file

Reviewer 3 Report

I have reviewed the paper entitled “Modeling of mangetospheres of terrestrial exoplanets in the habitable zone around G-type stars” by E.S. Belenkaya, I.I. Alexeev and M.S. Blokhina.

This paper discusses models of magnetospheres of planets within the habitable zone, impacted by the magnetic field of stellar winds and how they are being deformed depending on the on the parameters of the system. The paper addresses an interesting astrophysical question and it suggests that the structure of the exoplanetary magnetosphere depends on the above mentioned parameters.

In its current form, the paper is not sufficiently clear for publication. In particular, the authors should address the following points in a revised version.

  1. The authors adopt a paraboloid model where the magnetopause is a paraboloid by revolution. This implies that the magnetopause is axisymmetric with respect to the star-planet axis. The magnetic field of the planet itself cannot be axisymmetric, which is evident from Figure 1, yet, there is no discussion at all of this asymmetry in the equations presented in the paper, as they only have a dependence on the radial distance. The field lines shown in the figure however, have a dependence on the meridional angle \theta, and presumably on the azimuthal angle \phi. The authors do not explain how these figures were made, to what equations the field corresponds to and overall the results are not reproducible.
  2. There are several things mentioned in the model such as a current system in the magnetotail and the magnetopause. It is not clear which equations describe these currents, how they are treated and their impact on the system.
  3. The authors call section 3, as "Numerical Results". Does this imply that they use some numerical model i.e. integrate a set of PDEs, or run a set of simulations? This is not clear, either so they need to clarify that.
  4. The quality of Figure 1 is not clear and the reader cannot see the scales or the field lines. A higher quality-resolution figure is needed.
  5. The conclusion section is only qualitative. The authors need to provide some quantitative results on the dependence and the sensitivity of r_{mp} on the magnetic field field and possibly other quantities.

Author Response

Please, see the attached file

Round 2

Reviewer 1 Report

The manuscript authors considered all observations of the foregoing review. Now, in my opinion, the manuscript can be recommended for publication in the journal “Universe”.

Reviewer 2 Report

Thank you to the authors for making the requested changes. With the added referencing and explanations, I feel the paper is now ready for publication.

Reviewer 3 Report

The authors have addressed the comments raised in my previous report. So, I am happy to recommend acceptance of the paper in the current form.